

# Discriminating fluvial fans and deltas: Channel network morphometrics reflect distinct formative processes

Luke Gezovich[1][*], Piret Plink-Björklund[1], Jack Henry[1,2]

[1] Colorado School of Mines, Geology & Geologic Engineering, 1500 Illinois Street, Golden, CO, 80401

[2] Rice University, Earth, Environmental and Planetary Sciences, 6100 Main St., Houston, TX, 77005-1827

*Correspondence to:* Luke Gezovich (lukegezovich@mines.edu)

**Abstract**

Recent recognition of a new type of fluvial system – fluvial fans – introduces a fan-shaped channel network that appears similar to that of river-dominated deltas. Deltas form where rivers enter lakes and oceans, while fluvial fans are terrestrial landforms. However, fluvial fans can reach the shorelines of oceans or lakes, and in such cases the distinction between fluvial fan and river-dominated delta channel networks become ambiguous. We currently lack fundamental understanding of these two landforms' morphometric differences, despite their high socioeconomic significance, vulnerability to natural hazards, and key differences in how these landforms respond to global climate change and urbanization. Here we review the relevant conceptual differences in delta and fluvial fan network morphodynamics, propose a set of quantitative morphometric criteria to distinguish fluvial fan and delta channel networks, and test these criteria on 40 deltas and 40 fluvial fans from across the world. This initial attempt to distinguish deltas and fluvial fans demonstrates that quantifying channel network angles, and trends in normalized channel widths and lengths provides efficient criteria, but some ambiguities remain that need to be resolved in future work. This research advances our mechanistic understanding of fluvial fan and delta channel networks and the recognition of modern and ancient landforms on Earth and other planetary bodies, such as Mars and Saturn's moon Titan.

**Plain Language Summary**

Fluvial fans are a newly recognized type of river system that look like river deltas, especially when they reach lakes or oceans. This study explores how to tell them apart by measuring the size and layout of channels in these fan-shaped landforms. Understanding these differences helps to predict how these landforms respond to climate change and urbanization, and identify them on Mars and other planetary bodies.



## 1. Introduction

River deltas are depositional landforms that form where rivers enter lakes or oceans. They are home to over half a billion people, host abundant and biodiverse ecosystems, and function as both economic and agricultural hubs (Saito et al., 2007; Tejedor et al., 2015). The form and function of deltas is intimately linked to the evolving structure of their channel networks that determine how deltas distribute sediment and nutrients (Passalacqua, 2017; Pearson et al., 2020; Tejedor et al., 2017). Delta channel network morphology results from an intricate balance between sediment erosion and deposition from river, tide, and wave energy fluxes. River fluxes create distributary channels and islands, tides roughen the shoreline and widen the channels, and waves smooth the shoreline and decrease the number of distributary channels (Broaddus et al., 2022; Galloway, 1975; Nienhuis et al., 2015, 2018; Paniagua-Arroyave & Nienhuis, 2024; Vulis et al., 2023). Deltas dominated by river energy fluxes (river-dominated deltas) (Galloway 1975; Nienhuis et al 2015; 2018; Broaddus et al 2022; Vulis et al 2023; Paniagua-Arroyave and Nienhuis 2025) characteristically form fan-shaped landforms with complex distributary channel networks (Fig. 1). In these deltas, channel network topology is defined by mouth bar deposition and consequent distributary channel bifurcation (Bates, 1953; Edmonds & Slingerland, 2007; Wright, 1977).

Fluvial fans are another type of fan-shaped landform with channel networks that share morphological similarities with the river-dominated delta channel networks (Fig. 2). Fluvial fans are a relatively newly acknowledged type of fluvial landform (Weissman et al., 2010; Ventra & Clarke, 2018), that forms via river avulsions or "channel jumps" across low-gradient floodplains (Chakraborty et al., 2010; Martin & Edmonds, 2023; North & Warwick, 2007). Rivers have been traditionally regarded as sediment transfer or bypass zones in source-to-sink systems (Allen, 2008; Fielding et al., 2012), whereas fluvial fans are net depositional and build significant stratigraphic thicknesses (Chakraborty et al., 2010; Moscariello, 2018; Weissmann et al., 2015). Fluvial fans are also called "wet" fluvial-dominated alluvial fans (Schumm, 1977), megafans (Singh et al., 1993), or distributive fluvial systems (DFS) (Weissman et al., 2010). Fluvial fans are distinct landforms from alluvial fans – which form by a combination of gravitational and streamflow processes, feature steep gradients (typically 2–12°), and have a relatively small radius typically less than 10 km (Blair & McPherson, 1994; Moscariello, 2018). Fluvial fans form some of the largest terrestrial landforms on Earth ($10^3$–$10^5$ km² in surface area) (Horton & Decelles, 2001; Leier et al., 2005) and have low gradients (typically 0.03–0.001°) (Brooke et al., 2022). Fluvial fans are abundant across Earth, and they form in diverse climatic and tectonic settings (Hartley et al., 2010; Weissman et al., 2010; Ventra & Clarke, 2018). Like deltas, fluvial fans are home to hundreds of millions of people, and these highly dynamic landforms are critical for their livelihood – supporting agriculture, fisheries, and freshwater access. For example, the Kosi fluvial fan experiences catastrophic river floods



that lead to large numbers of casualties and displaced populations (Sinha, 2009; Syvitski & Brakenridge,
2013), but also provides water and nutrients contributing to agricultural productivity and the overall
health of the ecosystem (Gupta et al., 2021).

While fluvial fans are terrestrial landforms, they can reach the shorelines of oceans (Fig. 2b) or lakes

(Figs. 2a, 2d and 2i). It is in such cases fluvial fan and river-dominated delta channel network distinction
becomes ambiguous, while wave-and tide-dominated deltas have distinctly recognizable morphologies
(Broaddus et al., 2022; Galloway, 1975; Nienhuis et al., 2015; 2018; Paniagua-Arroyave & Nienhuis,
2024; Vulis et al., 2023). We currently lack quantitative morphometric criteria for distinguishing river-
dominated delta and fluvial fan channel networks, despite their socioeconomic significance, key
differences in their natural hazard vulnerabilities, and in how they respond to global change. Deltas are
global change hotspots highly vulnerable to urbanization and climate change which can aggravate coastal
hazards and cause sea level rise (e.g., Syvitski et al., 2009; Giosan et al., 2014), and reduce sediment
supply due to river damming and artificial levees causing the drowning of deltas (e.g., Blum & Roberts,
2009; Syvitski et al., 2009; Giosan et al., 2014; Nienhuis et al., 2020; Paola et al., 2011).

Numerous fan-shaped landforms with channel networks have also been identified on other

planetary bodies such as Mars (Ori et al., 2000; Wood, 2006; Malin & Edgett, 2015) and Saturn's moon
Titan (Wall et al., 2010; Witek & Czechowski, 2015; Radebaugh et al., 2018). Deltas on planetary bodies
are important indicators of paleo-shorelines and have been utilized to reconstruct the shorelines and water
levels of ancient lakes and oceans on Mars (di Achille & Hynek, 2010). However, Martian paleo-ocean
shoreline reconstructions have so far yielded mixed results (De Toffoli et al., 2021). This discrepancy
could perhaps arise because shoreline-bound deltas have not been effectively distinguished from fluvial
fans on Mars, which may form thousands of kilometers inland from shorelines (Bramble et al., 2019;
Limaye et al., 2023; Tebolt & Goudge, 2022). Deltas also offer attractive targets for mission sites in
search of life due to their habitability and high biosignature preservation potential, as exemplified by the
selection of Jezero Crater for NASA's *Perseverance* rover, *Ingenuity* helicopter, and future Mars Sample
Return mission (Farley et al., 2020). Distinguishing deltaic and fluvial fan paleo-channel networks on
other planetary bodies is even more ambiguous, especially if the lakes and oceans are no longer present.

Over time, the accumulation of biogenic and sedimentary materials distributed via channel networks

contributes to the construction of stratigraphy. Fluvial fans and deltas are net depositional systems, as
both are characterized by spatially diminishing water surface slopes that reduce sediment transport
capacity, thereby producing spatiotemporal convergence and deposition of sediment (Ganti et al., 2014).
Consequently, in addition to their socioeconomic significance, both landforms significantly contribute to
the stratigraphic record, and their deposits can be used to decipher past environmental conditions. High
deposition rates in fluvial fans and deltas promote the preservation of environmental change signals in the



sedimentary record (Trampush & Hajek, 2017). Similar to modern river-dominated deltas and fluvial
fans, we lack morphometric criteria to distinguish these two fan-shaped channel networks in the
sedimentary record, such as in seismic datasets.
This study is motivated by developing quantitative morphometric distinction criteria for fluvial fan
and river-dominated delta channel networks. Prior work has established quantitative morphological
criteria for describing deltaic channel networks and linked these characteristics to theory (Chen et al.,
2021; Coffey & Shaw, 2017; Edmonds et al., 2011; Edmonds & Slingerland, 2007; Fagherazzi et al.,
2015; Ke et al., 2019; Passalacqua, 2017; Pearson et al., 2020; Tejedor et al., 2015, 2017). However, there
are no existing quantitative criteria to characterize fluvial fan channel networks or to differentiate the two
landforms. To develop such criteria, we review the relevant conceptual differences in delta and fluvial fan
network morphodynamics, propose quantitative morphometric criteria to distinguish fluvial fan and delta
channel networks, and test these criteria on 40 deltas and 40 fluvial fans (Supplementary Data) from
across the globe (Fig. 3). We test the robustness of the approach by analyzing differences in channel
network morphometrics concerning the size and gradient of the systems, hydroclimate conditions, lake
versus ocean terminations and tide- versus wave-influences in deltas, and channel morphology in fluvial
fans. We assess how effectively the proposed methods distinguish fluvial fans from river-dominated
deltas and examine why this distinction matters under global change. This work serves to improve our
mechanistic understanding of fluvial fan and delta evolution, and their accurate recognition on Earth,
other planetary bodies and in the sedimentary record.

**2.  Delta and Fluvial Fan Channel Network Morphodynamics**
The nature of channel networks is dependent on distinct morphodynamic processes responsible for
their formation (Edmonds & Slingerland, 2007; Fagherazzi et al., 2015; Tejedor et al., 2015). Below we
analyze differences in delta and fluvial fan morphodynamics and review existing morphometric criteria
for quantifying deltaic distributary channel networks. Our review is not comprehensive; rather, it focuses
on the specific processes which govern the formation of the morphometric characteristics that we can then
use for distinction of these two landforms, namely channel network angles, and downstream changes in
channel widths and lengths. There are other important characteristics of deltaic channel networks, linked
to water and sediment discharge distribution, entropy, and connectivity (Chen et al., 2021; Ke et al., 2019;
Passalacqua, 2017; Pearson et al., 2020; Tejedor et al., 2015, 2017). These aspects are not considered in
this review, because they are outside the scope of this study that seeks to distinguish deltaic and fluvial
fan channel networks using easily applicable morphometric criteria that can be used to both deltaic and
fluvial fan networks.



We use the terms bifurcation and avulsion as *processes* rather than a geomorphological feature of
channel splitting. *Bifurcation* is the process of channel splitting driven by mouth bar formation (Edmonds
& Slingerland, 2007). *Avulsions* are channel "jumps", where a channel changes its course due to channel
super-elevation or a more favorable (steeper) gradient at channel flanks (Gearon et al., 2024). Partial
avulsions split channels; however the process is distinct from *bifurcation* around a mouth bar.
**2.1 River Deltas**

Figure 1: Examples of delta channel networks: (a) Apalachicola, (b) Selenga, (c) Yukon, (d) Kobuk, (e) Poyang Lake, (f) Parana (g) Saskatchewan, (h) Mamawi lake, (i) Slave deltas. The colors indicate channel hierarchy (see Methods). Base imagery from Esri's World Imagery basemap (© Esri, Maxar, Earthstar Geographics, and the GIS User Community).



144   Deltas (Fig. 1) always form where the mouth of a river enters a standing body of water. Here, the

145  transport capacity of the turbulent jet decreases, and the "parent" stream jet flow experiences both lateral

146  and bed friction, causing the flow to decelerate and rapidly expand laterally (Bates, 1953; Wright, 1977;

147  Edmonds & Slingerland, 2007; Jerolmack & Swenson, 2007). As a result, the transport capacity of the

148  turbulent jet decreases and sediment is deposited as a mouth bar basinward of the river mouth (Edmonds

149  & Slingerland, 2007). The process of mouth bar deposition and growth eventually leads to the bifurcation,

150  or downstream branching of a single (parent) channel into two daughter channels (Axelsson, 1967;

151  Coffey & Shaw, 2017; Edmonds & Slingerland, 2007) (Fig. 4a). These daughter channels are separated

152  by an island or shallow bay where sediment transport is significantly reduced or nonexistent, and flow is

153  unchannelized (Coffey & Shaw, 2017). Mouth bar deposition and resultant channel bifurcation repeat

154  multiple times leading to the seaward advancement of the shoreline and the construction of a delta

155  distributary channel network (Olariu & Bhattacharya, 2006; Edmonds & Slingerland, 2007) (Fig. 4a).

156   Deltas also experience channel avulsions at the lobe-level (Slingerland & Smith, 2004). These deltaic

157  avulsions occur within a region of high-water surface slope variability caused by backwater

158  hydrodynamics that are characterized by spatial flow deceleration and deposition during low flows, and

159  flow acceleration and bed scour with high flows (Lamb et al., 2012; Chatanantavet & Lamb, 2014). As

160  the backwater zone sets the location for avulsion in deltas (Chatanantavet et al., 2012), they are strongly

161  controlled by hydrodynamics in their receiving basin like bifurcations. As a result, the delta lobe size is

162  generally consistent and the lobe avulsion node migrates downstream commensurate with shoreline

163  progradation (Ganti et al., 2014). These avulsions episodically rearrange the depocenter at the delta lobe

164  scale, whereas the substantially more frequent bifurcations generate the topology of the delta distributary

165  channel networks (Edmonds & Slingerland, 2007; Bentley et al., 2016).

166   Resultant delta channel networks have a specific angle at which distributary channels bifurcate (Fig.

167  4a) (Coffey & Shaw, 2017), because a channel bifurcation will grow toward an equilibrium angle of 72°

168  to maximize flux at the two channel tips (Coffey & Shaw, 2017; Devauchelle et al., 2012; Ke et al., 2019;

169  Mahon et al., 2024). First described in tributary networks, this theoretical angle arises from diffusive

170  groundwater flow (Devauchelle et al., 2012). Testing of this concept reports bifurcation angles of 70.4° ±

171  2.6° (n = 9) in natural deltas (Coffey & Shaw, 2017), and 68.3° ± 8.7° (n = 21) (Coffey & Shaw, 2017)

172  and 74.1° ± 7.7°; (n = 13) (Federici & Paola, 2003) in experimental deltas.

173   The deltaic channel networks tend to consistently self-organize (Fagherazzi et al., 2008; Edmonds et

174  al., 2011) and exhibit a theoretical fractal pattern of decreasing channel widths and lengths associated

175  with increasing bifurcation order (Edmonds et al., 2011; Edmonds & Slingerland, 2007; Hariharan et al.,

176  2022; Seybold et al., 2017; Wolinsky et al., 2010) (Fig. 4a). The channel width trends align with

177  hydraulic geometric scaling: as the discharge of a parent channel divides into the discharge for two



resultant daughter channels, the daughter channel dimensions decrease as they scale with bankfull
discharge (Edmonds & Slingerland, 2007). Channel lengths decrease downstream with each successive
bifurcation because the jet momentum flux and consequent average grain transport distance decrease
downstream, causing new mouth bar deposition and accompanying bifurcations to occur closer to the
previous bifurcation node for a given channel (Edmonds & Slingerland, 2007) (Figs. 4a and 5a).
The nature of delta channel networks is further affected by waves and tides (Jerolmack & Swenson,
2007; Geleynse et al., 2011; Broaddus et al., 2022), where the relative strength of river, wave, and tide
processes determines whether deltas are river, wave, or tide dominated (Galloway, 1975; Nienhuis et al
2015, 2018; Nienhuis et al., 2020a; Vulis et al 2023; Paniagua-Arroyave and Nienhuis, 2025). Since wave
and tide-*dominated* deltas exhibit distinct morphologies from river-dominated delta and fluvial fan
channel networks, they are not considered in this study.
**2.2 Fluvial Fans**
In contrast to deltas where bifurcations and avulsions are strongly controlled by hydrodynamics
near a receiving basin of standing water (Chatanantavet et al., 2012; Ganti et al., 2014), fluvial fan river
avulsions are driven by a topographic slope break (Ganti et al., 2014; Martin and Edmonds, 2023).
Increased likelihood of avulsions at the fan apex is a consequence of the gradient reduction that triggers
in-channel sediment aggradation (Parker et al., 1998). These avulsions result from high channel bed
aggradation rates that are considerably higher than on the surrounding floodplains (Pizzuto, 1987). This
process causes river channel superelevation which ultimately triggers river avulsions near the fan apex
(Bryant et al., 1995; Mohrig et al., 2000; Gearon et al 2024). Since this slope break controls the location
of the fluvial fan's apex, the avulsion node is thus topographically pinned (Ganti et al., 2014). Partial or
full avulsions do occur further downfan, involving local gradient or discharge decreases, or crevassing
processes (Assine, 2005; Chakraborty et al., 2010; Donselaar et al., 2013; Gearon et al 2024) (Fig. 2).
Fluvial fan channel networks result from repeated avulsions that superimpose new channel positions on
paleo-channel locations and split channels by partial avulsions and crevasses. This generates apparent
channel "bifurcations" (North & Warwick, 2007) (Fig. 4b). However, as a process these are not
bifurcations related to mouth bar deposition but rather generated by avulsions. Fluvial fan channel
networks are predominantly paleochannel networks rather than active channel networks like in deltas
(North & Warwick, 2007; Chakraborty et al., 2010). Multiple channels can actively transmit discharge at
partial avulsions, such as during major river floods.






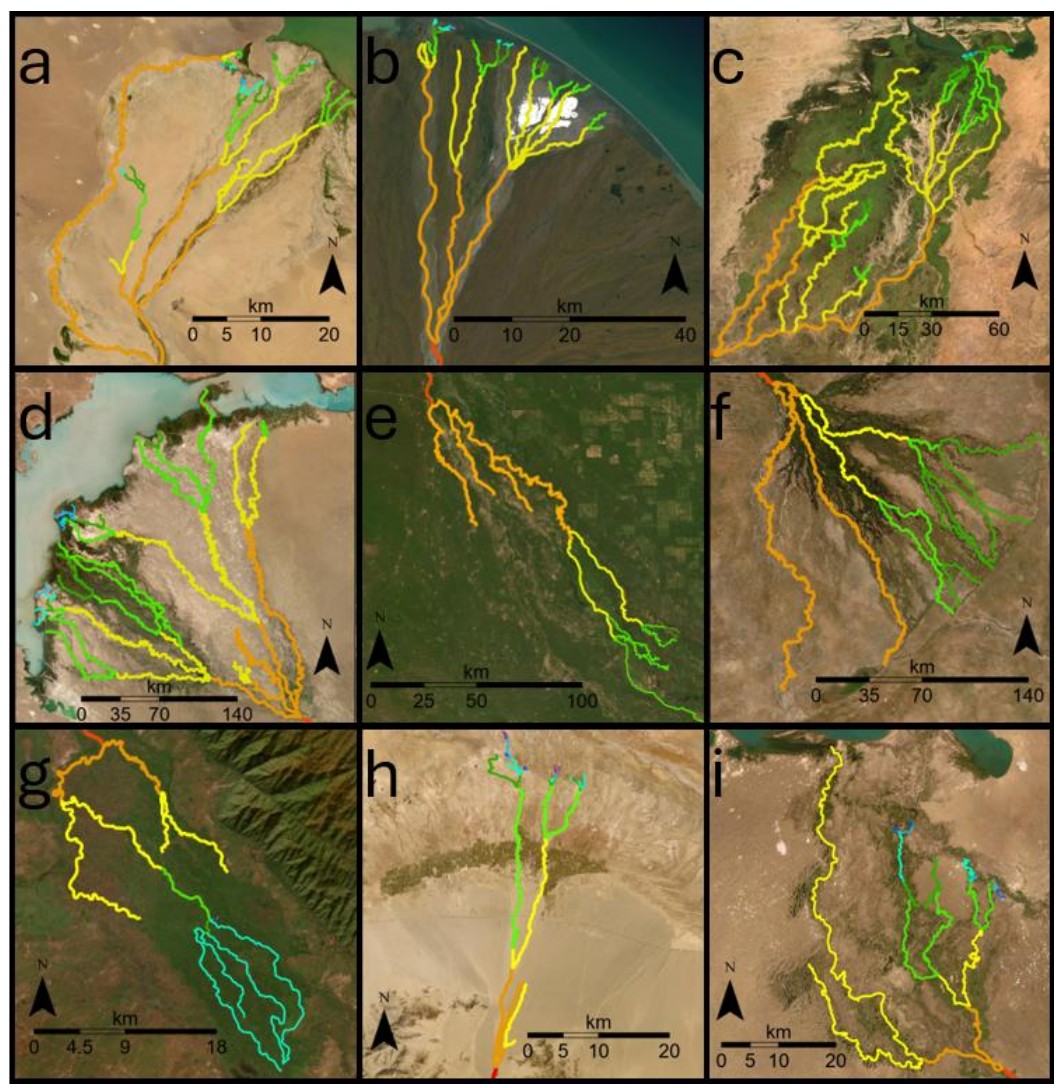

Figure 2: Examples of fluvial fan channel networks: (a) Dzavhan Gol, (b) Kongakut, (c) Niger, (d)
Ili, (e) Pilcomayo, (f) Okavango, (g) Shire, (h) Nomon He, and (i) Aksu fans. The colors indicate
channel hierarchy (see Methods). Base imagery from Esri's World Imagery basemap (© Esri,
Maxar, Earthstar Geographics, and the GIS User Community).

Downstream decrease in channel width has been documented in modern and ancient fluvial fans
(Nichols, 1987; Kelly & Olsen, 1993; Nichols & Fisher, 2007; Weissman et al., 2010; Davidson et al.,
2013; Owen et al., 2015; Wang & Plink-Björklund, 2019), linked to discharge losses to floodplain
processes, infiltration into the loose sediments of the fan, and evapotranspiration (Horton & Decelles,
2001; Hartley et al., 2010; Weissman et al., 2010; Davidson et al., 2013). However, some fluvial fan



channels have also been shown to widen downstream, possibly due to changes in channel planform or
aspect ratio, discharge contribution from groundwater, or discharge capture from adjacent rivers
(Chakraborty et al., 2010; Davidson et al., 2013). Fluvial fan channel networks have been studied for
qualitative descriptions of channel planform morphology (Davidson et al., 2013; Hartley et al., 2010;
Weissman et al., 2010), and scaling relationships (Davidson et al., 2013; Davidson & Hartley, 2014).
Modeling establishes a relationship between the fluvial fan shape and avulsion dynamics, such as
avulsion trigger period and abandoned channel dynamics (Edmonds et al., 2022; Martin & Edmonds,
2023).

Fluvial fans are distinct landforms from alluvial fans that feature steep gradients (typically 2–
12°), have a relatively small radial distance typically less than 10 kilometers, and lack channel networks
(Blair & McPherson, 1994; Moscariello, 2018). Although surface channels may occur on alluvial fans,
these are transient features formed by surface erosion, and do not construct alluvial fans which form by a
combination of gravitational and sheet flood processes (Blair & McPherson, 1994; Moscariello, 2018).
Thus, alluvial fans are not considered here as they are distinct from fluvial fan channel networks that form
by river avulsions.
**2.3 Morphometric Criteria for Recognition of Delta and Fluvial Fan Channel Networks**
Based on the above differences in delta and fluvial fan morphodynamics, we hypothesize that the
morphometric differences in their channel networks can be quantified. Based on prior work, we expect
river-dominated delta channel networks to display downstream decreasing channel widths and lengths
with increasing bifurcation order (Edmonds & Slingerland, 2007; Seybold et al., 2007; Wolinsky et al.,
2010), and have an average channel network angle of approximately 72° (Coffey & Shaw, 2017). These
metrics should differ in fluvial fans, because the channel networks are built by avulsions rather than
bifurcations. However, also delta networks experience avulsions and we expect some overlap in the
network angles. Below, we test these morphometric criteria on 40 river-dominated delta and 40 fluvial fan
channel networks (Fig. 3).

**3. Dataset and Methods**

Although automated channel mapping tools like ChannelExtractor in TopoToolbox (Schwanghart
& Kuhn, 2010) and Rivamap (Isikdogan et al., 2017) exist, these existing methods rely on either terrain-
based flow routing or the detection of active surface water – typically  based on spectral characteristics –
to delineate river channels. However, fluvial fan channel networks are predominantly composed of
paleochannels that lack both clear topographic expression and surface water signatures. Both delta and
fluvial fan channels can also be only a few meters wide, often falling below the spatial resolution of
commonly available DEMs and remote sensing imagery. In such settings, the coarse resolution and



smoothing of subtle terrain in DEMs, especially in low-relief environments, further limit the effectiveness
of automated extraction. As a result, we are constrained to manual digitization, as described below.

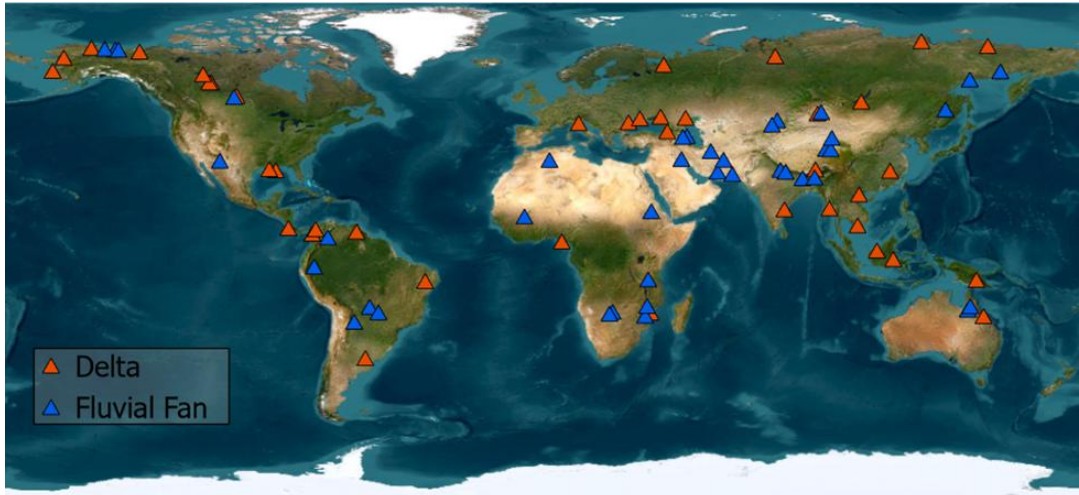

Figure 3: Map of deltas and fluvial fans in this study. Base imagery from Esri's World Imagery
basemap (© Esri, Maxar, Earthstar Geographics, and the GIS User Community).

### 3.1 Channel Order

To establish channel order in networks, we follow Dong et al. (2016). Their method follows a

simple rule: bifurcations produce downstream increasing channel order through channels that branch. To
be considered a channel of a higher order, the resultant channels must not merge downstream. When a
first-order channel bifurcates, two second-order channels develop downstream of this bifurcation. When
these two channels subsequently bifurcate, two new pairs of third-order channels form, and so on (Figs.
4a and 4b). Identification of bifurcation nodes follows Edmonds et al. (2011), such that the first-order
bifurcation for a river channel is the first bifurcation that the channel undergoes (Fig. 4a). Although these
methods were developed for deltaic channel networks, here we adapt them for fluvial fan networks also
(Figs. 4c and 4d). We do not consider channels that loop or rejoin downstream, or channels of non-fluvial
origin, such as tidal channels or inlets (e.g., Smart, 1971; Tejedor et al., 2015) that are not connected to
the fluvial distributary channels.

### 3.2 Channel Length and Width Measurements

Channel length and width measurements follow Edmonds and Slingerland (2007), where channel

length is measured as the distance between two bifurcation nodes in deltas (Fig. 4a). We adopt this
methodology also to fluvial fans to measure channel lengths between avulsion nodes (Fig. 4c). The
average width of a channel segment is recorded from three separate width measurements: one
immediately after a node ($w_i$), one immediately before the next node ($w_f$), and one halfway between these
two points at the midpoint of the channel segment ($w_h$) (Figs. 4a and 4c). Channel width measurements



were not performed in locations where a channel has locally split into multiple branches that join
downstream. In deltas, channel width measurements were recorded based on the width of water present in
the channel, as observed in the satellite imagery. For fluvial fans, paleo-channel width measurements
were based on the bankfull width, defined by clearly visible channel banks or vegetation boundaries. All
channel length and width measurements were normalized using the initial first-order channel width,

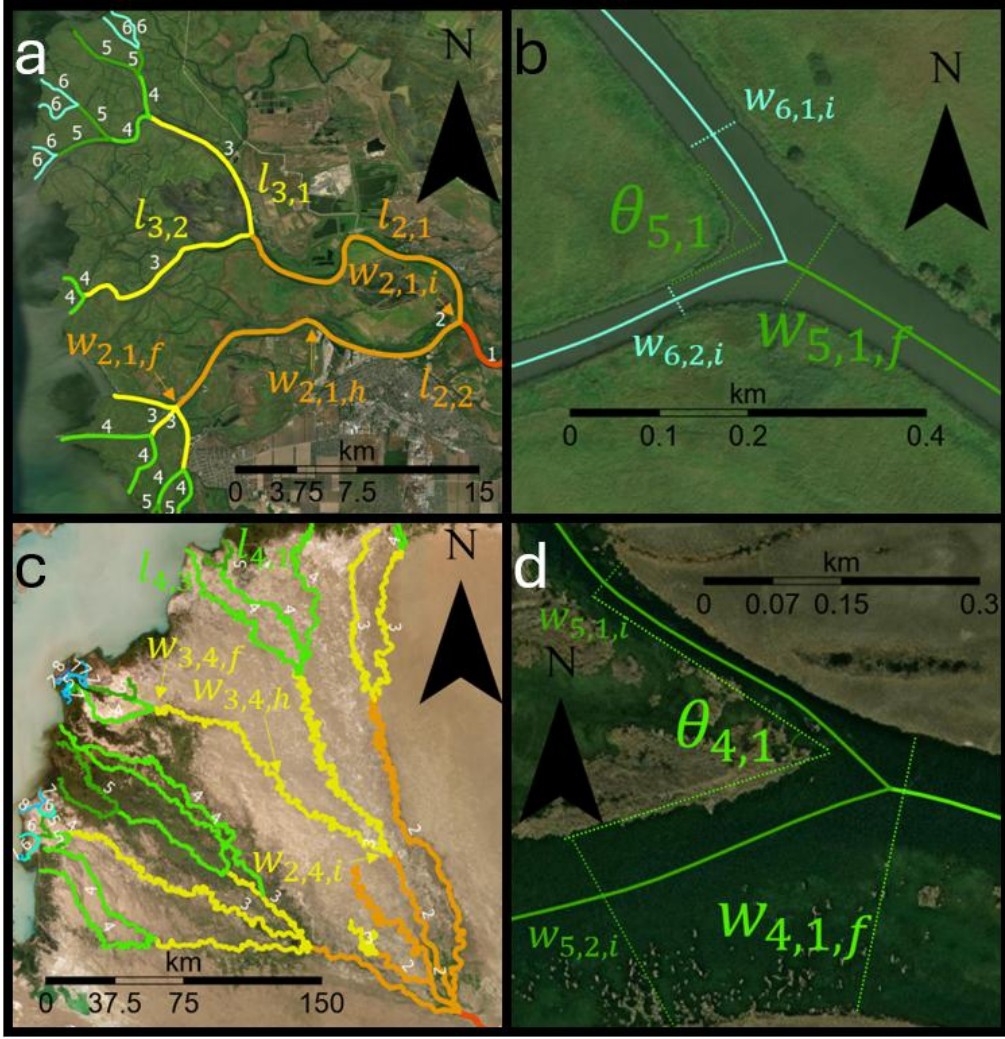

Figure 4: Illustration of (a) channel order, length, and width and (b) bifurcation angle measurements in deltas (Don delta). Illustration of (c) channel order, length, and width and (d) divergence/crossover angle measurement (Ili fan). Arrows point to locations of $w_i$ = initial channel width, $w_h$ = midpoint channel width, $w_f$ = final width measurements. The $w_f$ is set as the length of two limbs that track along the edges of the mouth bar. $\theta_n$ corresponds to the bifurcation or divergence/crossover order. Base imagery from Esri's World Imagery basemap (© Esri, Maxar, Earthstar Geographics, and the GIS User Community).



following the methodology of Edmonds & Slingerland (2007). Consequently, the normalized channel
width value for first-order channels is always equal to one. First-order channel lengths were measured
between the last occurrence of tributary channels and the first channel splitting node and contain no
significant value for our study.

**3.3 Network Angle Measurements**

To quantify network angles, we adopt the methodology of (Coffey & Shaw, 2017) developed for
measuring channel bifurcation angles, which determines the angles of mouth bars formed at the end of an
upstream channel. In this methodology, the final channel width directly upstream of a bifurcation ($w_f$) is
set as the length for two limbs of an angle that follows the mouth bar-water contact to measure a
bifurcation angle ($\theta_n$) (Coffey & Shaw, 2017) (Fig. 4b). The same methodology is adapted here for fluvial
fans (Fig. 4d). In some river deltas, tidal processes cause bifurcation of a channel into three channels
instead of two; these are referred to as trifurcations (Leonardi et al., 2013), and a few such measurements
are included in the dataset in the very distal portions of deltas where tidal influence is significant. We do
not measure angles where channels loop or rejoin downstream of avulsions or bifurcations. In essence, we
focus on the morphology of branching channel networks and measure the visible angles between channels
or paleo-channels independent of their origin (Fig. 4b and 4d).

**3.4 Global Delta and Fluvial Fan Channel Network Database**

To test the applicability of the proposed criteria, we selected 40 river-dominated deltas and 40
fluvial fans (Fig. 3 and Supplementary Data). These landforms were selected from a diverse range of
hydroclimatic, topographic, and basinal conditions from across the world (Fig. 3). All deltas have been
identified as such by prior work (Broaddus et al., 2022; Galloway, 1975; Hartley et al., 2010; Leier et al.,
2005; Nienhuis et al., 2015, 2018; Vulis et al., 2023). The river dominance of deltas and the presence of
tide- or wave-influence was determined using the established principles of process-based delta
classification (Broaddus et al., 2022; Galloway, 1975; Nienhuis et al., 2015, 2018; Paniagua-Arroyave
and Nienhuis 2025; Vulis et al., 2023). All included deltas display active discharge based on satellite
imagery. Only river-dominated deltas are included in the dataset, because wave-and tide-dominated delta
morphology is distinct from that of fluvial fans. Many natural river-dominated deltas are, however, tide-
or wave-influenced to varying extents. We test the effects of tide- and wave-influence on the
morphometric criteria by comparative analyses. Fluvial fans were located using their apex coordinates
from the global fluvial fan database of Hartley et al. (2010). This database also includes data on fluvial
fan length, gradient, termination style, such as axial, contributory, lacustrine, marine, playa, desert/dune,
and wetland styles. Contributory termination refers to that a distributive paleo-channel pattern becomes
contributory at the fan toe, and axial to fans where the active channels form a confluence with another
river (Hartley et al., 2010). We also subdivided delta termination styles in lakes and oceans. To test the





robustness of our methodology, we analyze whether the landform size, gradient, termination style, or
wave- and tide-influence in deltas affect the results.
**3.5 Mapping with ArcGIS Pro**
Delta and fluvial fan channel networks were mapped using ArcGIS Pro software (Version 3.2.1)
(Fig. 1, 2, and 4). Two feature classes were created: one for deltas and one for fluvial fans. Each delta or
fluvial fan landform was then individually mapped as a shapefile layer under the corresponding feature
class. The shapefiles for channel networks were created as polyline features, which allow a user to
manually trace individual river channel segments while automatically recording line lengths. Channels
widths and angles were measured using the line and angle measurement tools in ArcGIS Pro. All data was
recorded in the attribute table for each landform. This data was then exported and organized into Excel
documents and subsequently converted to a python and pandas readable CSV files (Supplementary Data).
A limitation of our methodology is the uncertainty regarding how soon satellite images were
captured after a precipitation event for a given landform, which can significantly influence channel
discharge and affect measured channel widths, especially in arid fluvial fans. Such events can also
reactivate partial avulsions and crevasse, potentially increasing the apparent number of channels.
However, none of the selected systems exhibited observable seasonal or significant discharge changes
across their channel networks. Additionally, because this study relies on values normalized to the initial
channel width, the effects of seasonal variability on channel width measurements are minimized.
**3.6 Code and Statistics**
Kolmogorov-Smirnov and Shapiro-Wilk tests were first applied to determine whether the data is
normally distributed. Levene's test was used to test for differences in variances in populations which do
not exhibit a normal distribution (Trauth, 2006). Independent samples or Welch's T-test were then applied
to test for a difference in means for populations with similar and dissimilar variances, respectively, while
one-sample T-tests were used to test comparisons of a subgroup against the overall population mean
(Trauth, 2006). For this study, a p-value less than 0.05 (5% significance level) suggests that the two
population distributions, variances, or means are not similar. Data analyses confidence intervals were
calculated according to Mendenhall et al., (2012). Data analysis and visualization were performed using
Python. Open-source data visualization libraries Matplotlib (Hunter, 2007), NumPy (Harris et al., 2020),
SciPy (Virtanen et al., 2020) and Seaborn (Waskom, 2021) were utilized.

**4.  Results**
**4.1 Delta and Fluvial Fan Channel Network Angles**
The average channel network angle ($\theta_d$) in deltas is 73.8° with a 95th percentile confidence interval of
± 1.9° (n = 528) (Fig. 5a). The average channel network angle ($\theta_f$) in fluvial fans is 55.0° ± 2.0° (n = 520)





(Fig. 5b). The delta and fluvial fan network angle populations are not normally distributed according to
both Kolmogorov-Smirnov (KS) and Shapiro-Wilk (SW) tests, with p-values less than 0.05. Levene's test
for statistical difference in variances also results in a p-value less than 0.05, suggesting population
variances are statistically different. A subsequent independent samples T-test suggests the means of delta



and fluvial fan angle populations are statistically different, with a p-value less than 0.05. All statistical
results are recorded in Supplementary Table 1 in the Supplementary Information.



Figure 5: Histograms depicting distributions of (a) delta angles with average delta angle ($\theta_d$), its standard deviation ($\sigma_d$) and median and (b) fluvial fan angles with average fan angle ($\theta_f$), its standard deviation ($\sigma_f$), and median displayed. Box-and-whisker plot displaying the average angle for each delta (c) and fluvial fan (d) landform ($\theta_{Landform}$).



We also reviewed the average network angle of each individual delta and fluvial fan ($\theta_{landform}$) (Figs.
5c and 5d), and these analyses reveal some overlap. All fluvial fans have average angle values less than
60°, except for six landforms, or 15% of fluvial fans in this study. Four of these landforms have average
angles larger than 60° (60.8°, 63.2°, 67.7°, 67.9°), and two larger than the delta average of 73.7° (79.6°,
80.1°). All individual deltas have average network angles larger than 60°, except for one delta (59.3°).
There are also three deltas with average angles around 60° (61.5°, 62.4°, 63.3°).
The distribution of delta angles grouped by order (Fig. 6a) yields no strong trends for mean angle in
deltas. Seventh and tenth order channels have slightly lower average angle values at 65° and 67°, but
these higher-order groups have low sample sizes (n = 3; n = 8) (Fig. 6a). The distribution of fluvial fan
angles grouped by order does yield a trend: the average angle for first through third orders ($\theta_1$, $\theta_2$, and $\theta_3$
in Fig. 6b) is between 47 – 50°, and increases to 61 – 63° for fourth through eight order channels, and to
66° for ninth order angles (n = 6) ($\theta_4 – \theta_9$, in Fig. 6b). In contrast to the unimodal distribution of delta
angles, the distribution of higher-order fluvial fan angles is bimodal, with a dominant peak near 50° and a
secondary peak around 80 – 100° (Fig. 6b).

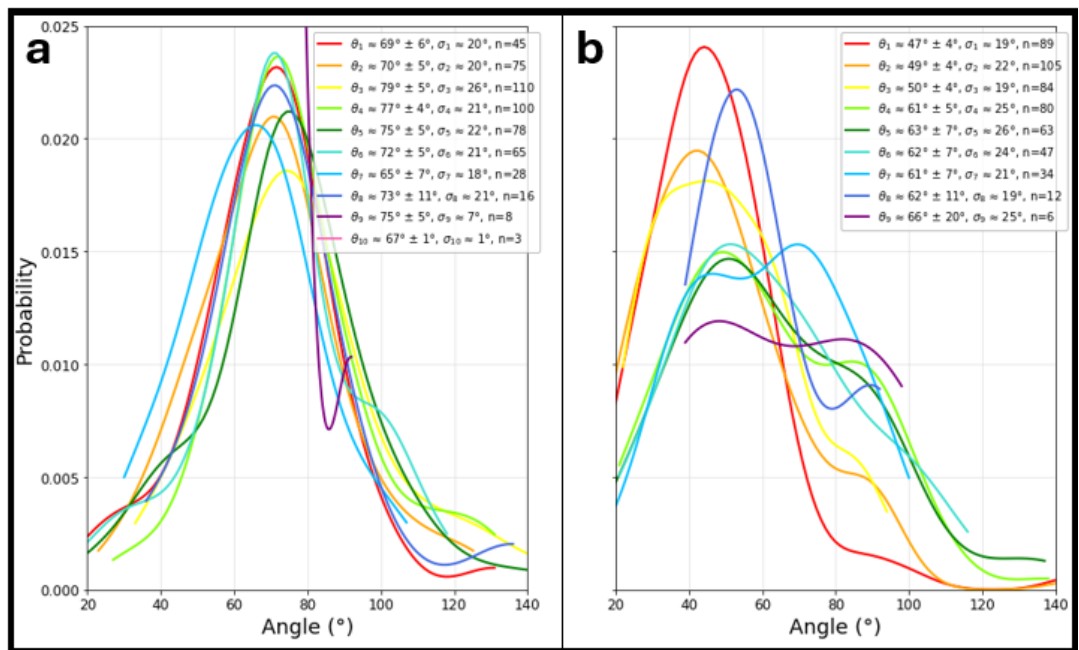

Figure 6: Distribution of (a) delta bifurcation angles, and (b) fluvial fan divergence/crossover angles grouped by order ($\theta_n$) with the 95th percent confidence interval. ($\sigma_n$) = denotes standard deviation. n denotes sample size.

All deltas in this analysis are river-dominated deltas, however some are tide- or wave-influenced.
Grouping deltas by process regime shows that the average bifurcation angle for the 19 river-dominated



deltas ($\theta_R$) is 73.4° ± 2.2  (n = 375), for the 16 tide-influenced deltas ($\theta_t$)  75.6° ± 3.9 (n = 139) and for the
5 wave-influenced deltas ($\theta_w$) 67.1° ± 10.1 (n = 14) (Fig. 7a). The river-dominated and tide-influenced
delta angle means are not statistically different from the mean angle for the whole delta population
(Supplementary Table 1). The wave-influenced delta angles were omitted from this statistical analysis due
to a small sample size (n = 14 < 30).

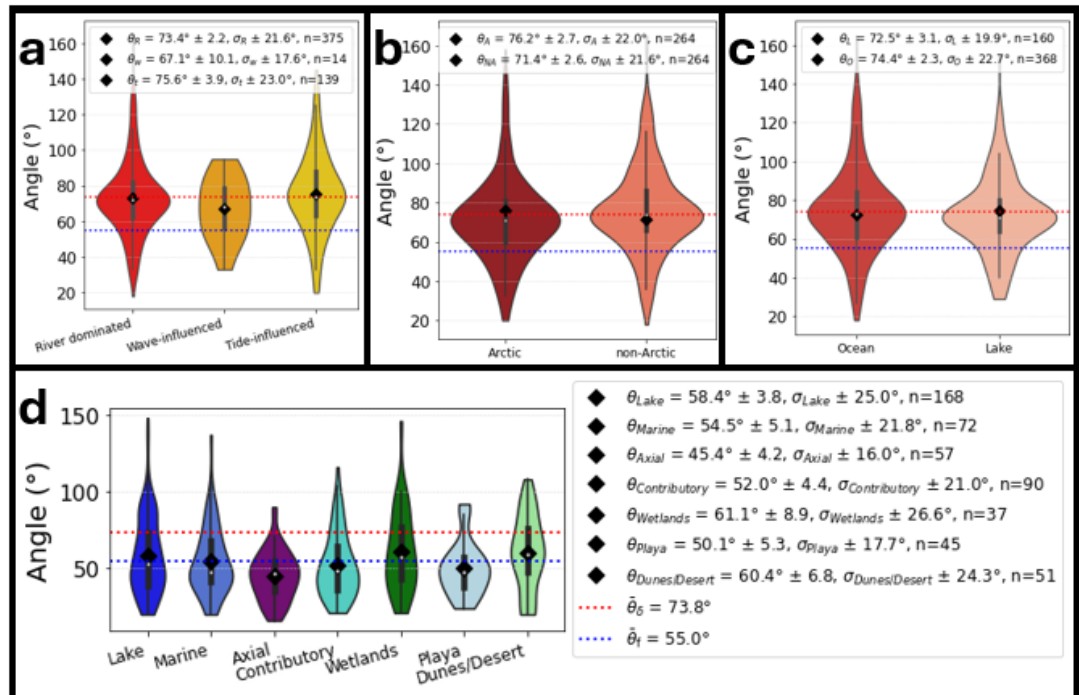

Figure 7: Violin plots depicting angle distributions for (a) delta process regime: river dominated ($\theta_R$),
wave-influenced ($\theta_w$), and tide-influenced ($\theta_t$), (b) deltas in non-Arctic ($\theta_{NA}$) and Arctic ($\theta_A$)
climates, (c) ocean terminated deltas ($\theta_O$) and lake terminating deltas ($\theta_L$), and (d) fluvial fan
termination styles. All average angle values have a corresponding 95th percent confidence intervals,
standard deviation ($\sigma$), and sample count (n).

Many delta angle measurements in this dataset come from arctic deltas. The comparison between

Arctic and non-Arctic deltas shows that Arctic deltas have a larger mean angle ($\theta_A$ = 76.2° ± 2.7, n =
264), than non-arctic deltas ($\theta_{NA}$ = 71.4 ± 2.6, n = 264) (Fig. 7b). There is a statistically significant
difference in means between Arctic and non-Arctic deltas (Supplementary Table 1). Grouping deltas by
termination style (Fig. 7c) shows that deltas which terminate in lakes have slightly smaller mean angles
than those that terminate in oceans ($\theta_L$ = 72.5° ± 2.7, n = 160 versus $\theta_O$ = 74.4° ± 2.6, n = 160), but these
differences are not statistically significant compared to the whole delta population (Supplementary Table

1).

Grouping fluvial fans by their termination style shows some differences (Fig. 7d), where the mean





angles vary from a low of $\theta_{Axial} = 45.4° \pm 4.2$ (n = 57) for axial-terminating fluvial fans to $\theta_{wetlands} = 61.1°$
$\pm 8.9$ (n = 37) for wetland terminating fans (Fig. 7d). All fluvial fan termination types, except for axial-
terminating fans, exhibit population means that are statistically similar to the overall fluvial fan
population (Supplementary Table 1). However, each termination style is represented by only 4 to 6
landforms, limiting the statistical power of comparisons and generalizations, despite the relatively robust
measurement numbers in wetland (n = 37), playa (n = 45), dunes/desert (n = 51), and axial-terminating
fans (n = 57). There also appears to be some discrepancies in Hartley et al. (2010) assignment of
termination types. We also tested whether landform size (Supplementary Fig. 1) and gradient
(Supplementary Fig. 2) affect the channel network angles, and these analyses yield no trends, supporting
the robustness of our methodology.
**4.2 Channel Lengths and Widths**

Normalized channel length and width measurements reveal morphological differences between

fluvial fan and delta channels. Both landform types show non-linear decreases in these values with
increasing channel order (Fig. 8). Statistical analyses confirm that the overall means for normalized
channel length and width differ significantly between fluvial fans and deltas (Supplementary Table 1).
Fluvial fan channels are generally an order of magnitude longer than delta channels, with mean
normalized length of 147.09, compared to 17.18 in deltas (Figs. 8a and 8c). In contrast, delta channels
tend to be slightly wider, with normalized mean width of 0.40 compared to 0.26 in fluvial fans (Figs. 8b
and 8d).

Comparing the normalized dimensions by channel order (Fig. 9) shows further trends. The lower

order normalized channel widths (orders 1–5) in fluvial fans are significantly longer, and the channel
shortening rate is higher compared to deltas (Fig. 9a). The normalized lengths become very similar in
orders 7–8, and then diverge again for the higher orders where the fluvial fan channel lengths are
somewhat longer, but the channel shortening rates are higher in deltas. Normalized channel widths show
significant differences for orders 2–8, but not for 9–11. Only a few landforms have channels with orders
exceeding 9. Fluvial fan narrowing rates are very high from order 1 and 2, and very low in orders 7–10
(Fig. 9b). The narrowing rates are more uniform in deltas.

When comparing individual deltas by process regime,  tide- and wave-influenced deltas have a

significantly higher mean normalized channel widths relative to the whole delta population
(Supplementary Fig. 3 and Supplementary Table 1).



Figure 8: Box and whisker plots illustrating normalized delta channel widths (a) and lengths (b) and normalized fluvial fan channel widths (c) and length (d), plotted by channel order.




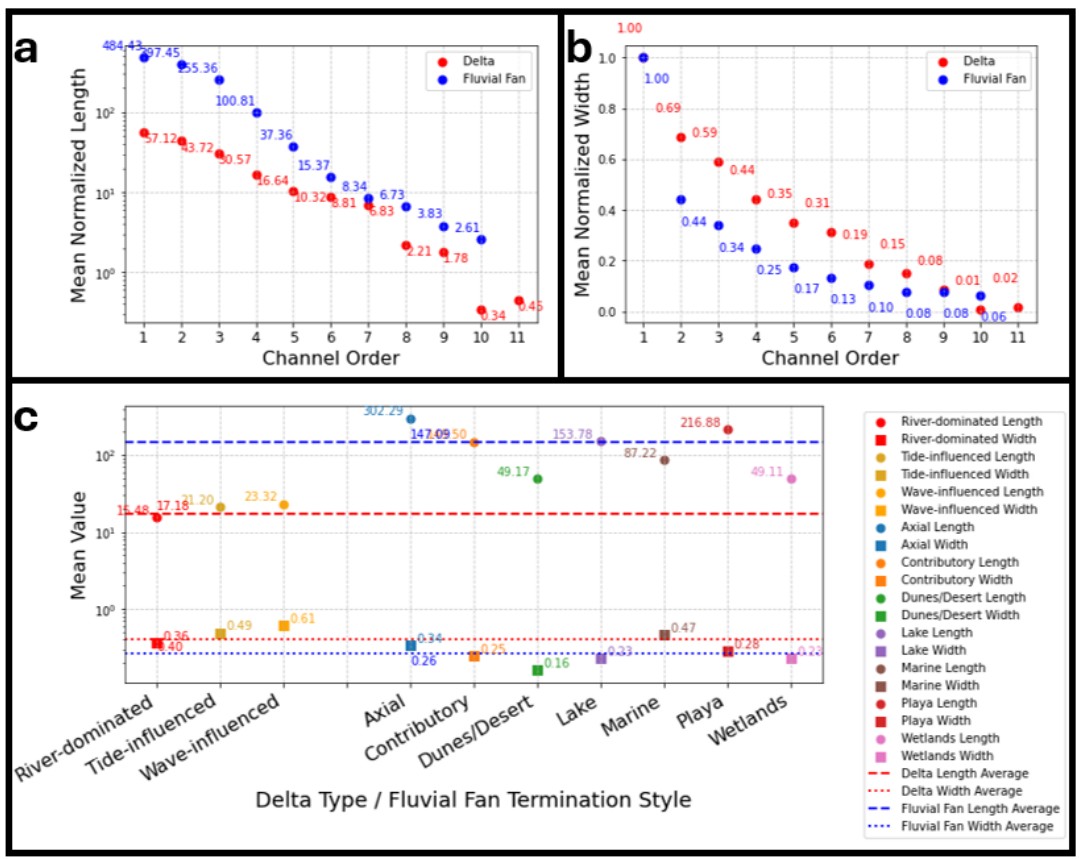

Figure 9: Mean normalized delta and fluvial fan channel (a) lengths by order and (b) width values by order. (c) Mean channel length and width values for different types of deltas and fluvial fan termination styles.

Comparison by fluvial fan termination styles, shows that axial and playa-terminating fans exhibit
longer mean normalized channel lengths compared to the whole fluvial fan population, whereas
dunes/desert, marine, and wetland-terminating fans have shorter mean lengths (Supplementary Fig. 3 and
Supplementary Table 1). Contributory and lake-terminating fans do not differ significantly from the
overall mean. Regarding normalized channel widths, axial and marine fans have wider channels, while
dunes/desert fans are narrower. Normalized width values for contributory, lake, playa, and wetland fan
channels show no difference from the whole population mean (Supplementary Fig. 3 and Supplementary
Table 1). Statistical analyses of channel length and width were not conducted for different fluvial fan
termination styles due to insufficient sample sizes (n < 30) in most categories.

**5. Discussion**



**5.1 Effectiveness of Morphometric Criteria in Distinguishing Deltas and Fluvial Fans**


The average channel network angles are distinctly different in deltas and fluvial fans by 20°, and
this statistically significant difference is a useful criterion in distinguishing these two landform types.
While some overlaps exist at the landform level, these cases are relatively limited, where 15% of fluvial
fans in this dataset have an average angle larger than 60° (Fig. 5d) and 10% of deltas have an average
angle less than 64° (Fig. 5c). These findings support the utility of average branching angle as a
distinguishing metric between deltas and fluvial fans, though some uncertainty remains, and additional
criteria are necessary for more robust distinction.
An additional criterion is the distribution of average angles by channel order, where fluvial fans
have increasing angles and a bimodal distribution in orders 4–8 (Fig. 6). Other supportive criteria may be
the differences in values and distributions of the normalized channel lengths and widths (Figs. 8 and 9),
but the low sample numbers do not allow us to test these criteria by individual landforms. A useful
criterion would be to link channel narrowing with the bifurcation and avulsion nodes. In deltas, the
downstream channel narrowing occurs in stepwise manner at the bifurcation nodes, whereas in fluvial
fans this decrease should be gradual and not linked to the node positions where full avulsions occur. Our
data was collected in a manner that does not allow us to do these analyses.
A potential source of overlap in the delta and fluvial fan channel network average angles is that
not all measured angles in deltas are bifurcation angles, as deltas also experience avulsions (e.g., Fig. 1e).
A closer inspection of the four deltas with low average network angles reveals that each contains very few
measurements ($n = 3$, $n = 4$, $n = 6$, $n = 7$). In these cases, the limited sample size allows the rarer avulsion
angles to affect the mean values more strongly.
Examining fluvial fans with high average angles shows that these are low-gradient wetland fans,
where the avulsion angles tend to be wider as a function of avulsion mechanisms (see Discussion below).
However, they may also suggest a methodological limitation. While the local avulsion angles in low-
gradient wetland fans are wide (measured the final channel width directly upstream of a bifurcation ($w_f$)
as the length for two limbs of an angle), angles between the longer channel reaches are considerably
narrower (Supplementary Fig. 4). We plan to further develop angle measurement methods to capture
both the local and the reach-scale angles in future work.
In summary, this initial attempt to distinguish deltas and fluvial fans demonstrates that
quantifying channel network angles, and trends in normalized channel widths and lengths provides
efficient criteria. However, we also show that sample sizes are important for accurate recognition of
landforms.
**5.2 Processes that determine delta and fluvial fan channel network angles**



While the 72° average bifurcation angle has a theoretical explanation in diffusion in non-channelized
flow (Coffey & Shaw, 2017), there is currently no established explanation for the approximately 55°
average network angle in fluvial fans. In deltas, bifurcation as a process is the product of sedimentation
from turbulent jets that form at the mouths of rivers entering basins (Bates, 1953; Coffey & Shaw, 2017;
Edmonds & Slingerland, 2007; Fagherazzi et al., 2015; Jerolmack & Swenson, 2007; Wright, 1977).
Once a mouth bar is formed, the flow through the distributary channel bifurcations can be modeled as
diffusive flow (Coffey & Shaw, 2017), and the resulting critical angle of 72° represents a stable
morphology for the bifurcation as it grows in a diffusive groundwater field (Devauchelle et al., 2012; Ke
et al., 2019). The slightly larger network angles in Arctic deltas may reflect environmental influences
such as ice cover, permafrost, or limitations on overbank flow (Lauzon et al., 2019; Piliouras et al., 2021;
Walker, 1998).
River avulsions are set up by channel superelevation (Mohrig et al., 2000), or when the slope down the
flanks of the channel provides a steeper descent than the existing river channel (Slingerland & Smith,
1998; Törnqvist & Bridge, 2002). Avulsions result from channel bed aggradation that reduces the channel
capacity (Bryant et al., 1995). Once an avulsion is triggered, and full or partial river flow exits the
channel, a new channel is generated by surface runoff erosion. Thus, the prevailing topographic gradient
would tend to keep the nearby flows more focused in a slope-parallel direction, compared to bifurcations,
resulting in narrower network angles compared to bifurcations (Fig. 5b).
The contrast between diffusion-dominated and surface runoff erosion-dominated processes in shaping
delta versus fluvial fan channel network topology is further supported by tributary channel network
analyses that originally defined the critical angle of 72° (Devauchelle et al., 2012). Tributary channel
network analyses show that the average tributary angle of 72° only occurs in humid catchments with high
groundwater recharge, where tributary networks are shaped by groundwater diffusion (Seybold et al.,
2017). In contrast, tributary network angles average at 45° in arid landscapes where surface runoff
dominates (Seybold et al., 2017), or are even lower in the driest catchments (Seybold et al., 2018).
Fluvial fan gradient decreases progressively downstream (e.g. Chakraborty et al., 2010), such that
higher gradients near the fan apex likely generate more acute angles, whereas the very low gradients near
the toe of the fan would allow for wider angles. This trend likely explains the downstream increase in
fluvial fan network angles and the emergence of the second, wider peak in higher order channels (Fig.
6b). Furthermore, avulsion mechanisms have been shown to change from channel superelevation in
upstream river reaches, where river gradients are steeper, to gradient advantage in downstream low-
gradient reaches (Gearon et al., 2024). In these low-gradient zones, crevassing processes can produce
high-angle deviations with the angle values around 90° (Rahman et al., 2022). Avulsion angles above
100° have been measured in meandering rivers on low-gradient floodplains with vegetation (see Rahman





et al., 2022). These effects may be important controls in the fluvial fan channel networks in low-gradient
vegetated wetlands. Reitz & Jerolmack, (2012) show that abandoned paleochannel reoccupation may
control new avulsion positions, and paleochannel density is highest in the narrower fan apex. Avulsion
angles may also change over time due to evolving channel width ratios (Morais & Montanher, 2022), or
may be affected by a critical angle or bend curvature (Yang, 2020).
We thus conclude that the distinction between deltaic and fluvial fan channel network angles arises
from the dominant formative processes: diffusive flow in deltas versus surface runoff erosion in fluvial
fans. Furthermore, in fluvial fans, network angles appear to be negatively correlated with surface
gradients, with lower gradients allowing for wider avulsion angles.

### 5.3 Ancient deltas and fluvial fans

Our proposed methodology could also be used to distinguish ancient fluvial fans and deltas, for
instance in seismic datasets, where only delta channel network angles have been quantified before
(Mahon et al., 2024). Our results confirm the prior modern data (Chakraborty et al., 2010) and recent
modeling outcomes (Martin & Edmonds, 2023), and help to eliminate a conundrum or discrepancy in
plan-view versus cross-sectional fluvial fan facies models (Plink-Björklund, 2021). Namely, earlier work
suggested bifurcations as a key mechanism driving fluvial fan formation (Friend, 1978; Kelly & Olsen,
1993; Weissman et al., 2010), probably due to downstream channel narrowing. However, this hypothesis
contradicts the stratigraphic data that indicate that proximal fans consist of amalgamated channel deposits
(Chakraborty et al., 2010; Kelly & Olsen, 1993; Nichols & Fisher, 2007; Singh et al., 1993; Weissman et
al., 2013) – a pattern consistent with frequent avulsions (Chakraborty et al., 2010; Singh et al., 1993).

### 5.4 Sensitivity of Deltas and Fluvial Fans to Global Change

Deltas and fluvial fans differ significantly in their vulnerability to natural hazards and in their
responses to global change. Deltas are highly vulnerable to coastal hazards and sea level rise (e.g.,
Syvitski et al., 2009; Giosan et al., 2014). Rising sea-level will not only inundate deltaic distributary
networks, but also cause a landward migration of the avulsion node corresponding with the landward shift
of the backwater zone (Chatanantavet et al., 2012; Ganti et al., 2014). This process reduces sediment
delivery to shorelines accelerating the effects of sea-level rise. In contrast, fluvial fans are controlled by
upstream morphodynamics, where the fan location (apex) is pinned by a topographic break (Ganti et al.,
2014; Martin & Edmonds, 2023). While sea-level rise and coastal erosion would affect the fan toes, the
avulsion node and fan apex position, and sediment delivery would not be affected, making fluvial fans
significantly less vulnerable to drowning.
Both deltas and fluvial fans are affected by reduced sediment supply due to river damming and
artificial levees (e.g., Blum & Roberts, 2009; Syvitski et al., 2009; Giosan et al., 2014; Nienhuis et al.,
2020; Paola et al., 2011). However, fluvial fans are highly sensitive to the water and sediment supply



changes, such as driven by changes in precipitation patterns (Leier et al., 2005; Assine et al., 2014;
Hansford & Plink-Björklund, 2020). Increases in extreme precipitation cause a significant increase in
avulsion frequency and crevassing splay formation (Morón et al., 2017), because large fluctuations in
river discharge, such as during extreme precipitation events, are avulsion-triggering events (Jones &
Schumm, 1999). Indeed, fluvial fans have been shown to be highly sensitive to such changes, where
fluvial fan activation and deactivation cycles have been linked to millennial-scale changes in monsoon
intensity or precipitation pattern (Assine et al., 2014; Fontana et al., 2014, Latrubesse et al., 2012).

**6. Conclusions**
This study demonstrates that river-dominated delta and fluvial fan channel networks can be
distinguished using quantitative morphometric criteria derived from their channel network topology.
Deltaic networks are primarily shaped by bifurcation processes resulting in average bifurcation angles of
approximately 74°, consistent with diffusion-dominated growth. In contrast, fluvial fan topology is
shaped by channel avulsions producing narrower average network angles near 55°, indicative of surface
runoff processes. Fluvial fan network angles tend to widen downstream, likely due to decreasing gradients
and avulsion style shifts, while delta angles remain relatively consistent, reflecting persistent bifurcation
processes. Both channel networks display downstream reductions in channel length and width with
increasing channel order, but the fluvial fan networks are characterized by significantly longer and
somewhat narrower channels when normalized.
These differences not only support the use of network morphology as a diagnostic tool for
identifying ancient fluvial fans and deltas in the stratigraphic record or other planetary bodies but also
provide insights into their differing sensitivities to environmental change.

**Code Availability**
The Python code used for data analysis and figure generation was created and run in Jupyter
Notebook version 6.4.8 (Anaconda distribution).

**Data Availability**
Morphological data collected in this study are available at https://github.com/lukegezovich/Delta-and-
Fluvial-Fan-Networks.

**Competing Interests**
The authors declare that they have no conflict of interest.



**Acknowledgments**
We thank reviewers Drs. Kamini Singha, Lesli Wood, and Wendy Zhou for their constructive
feedback that helped improve earlier versions of this manuscript.

**Financial Support**
Luke Gezovich thanks the American Association of Petroleum Geologists (AAPG) Foundation
John & Erika Lockridge Grant, the American Institute of Professional Geologists (AIPG) William J. Siok
Graduate Scholarship, the Colorado Scientific Society (CSS), the Rocky Mountain Association of
Geologists (RMAG), and the Society for Sediment Geology (SEPM) for providing funding to support this
research.



















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
