# Peer review of "Discriminating fluvial fans and deltas: Channel"

_EGUsphere, 2025_

## Referee Comment (RC2)

**Peer review of "Discriminating fluvial fans and deltas: Channel network morphometrics reflect distinct formative processes"**

This manuscript submitted to *Earth Surface Dynamics* is an interesting paper that uses satellite imagery to quantify the difference in bifurcation angle and channel dimensions between river deltas and fluvial fans. The authors find that bifurcation angle is lower in fluvial fans (n=40 fans) than river deltas (n = 40 deltas), which suggests that the differences in bifurcation process between those two landforms has a morphometric impact. Overall, this paper is clearly written with clear methodology and well-supported conclusions. However, I have some questions about study site selection that need to be clarified in the text (or perhaps adjusted with some supplemental study sites) before publication. Below I outline overall concerns/questions and then line-specific comments.

Main concerns & questions

*River deltas versus water-terminating fluvial fans*

The distinction between river deltas and fluvial fans is key to the premise of this paper, which asserts that these are two fundamentally different landforms with different bifurcation processes and thus different morphologies. Overall this makes sense to me, except in the case that fluvial fans terminate in standing bodies of water (i.e., oceans and lakes). If the distributary fluvial network ends in a standing body of water, isn't that simply a river delta? Lines 74-84 mention this ambiguity, but I don't understand how this is resolved in the paper. At a minimum, more text is needed to explain how some landforms that end in water are classified as fluvial fans, and others are classified as river deltas. Figure 7d shows the different terminations for the fluvial fans in this study (although not the n for each one), and many of them are in the "lake" and "marine" categories. How are those distinct from river deltas?

Addressing this question may be a matter of revision to the writing to better explain in advance how water-terminating fans differ from river deltas (i.e., spell this out clearly in line 74-84, using criteria that aren't the ones being tested in this study (bifurcation angle, etc.)). However, if such independent criteria don't exist, then I wonder if it is necessary to remove the lake- and marine-terminating fans from this analysis. Having two independent populations is very important in this comparative analysis, so I consider this to be a critical issue that needs to be addressed before publication.

*River vs wave vs tide delta criteria*

The matter of defining the type of delta (river vs wave vs tide) is unclear in this manuscript and needs clarification or to be refined with more quantitative criteria, if no quantitative metrics were applied before sites were chosen. In the intro (lines 187-188), the authors specify that only river-dominated deltas were used, but it is unclear how this is established and also there are results from wave- and tide-dominated deltas later on (e.g., Figure 9). The current methods sentence (lines 298-299) is definitely not specific enough about distinguishing between types. I do think the cited literature in this sentence is the appropriate body of work to establish specific criteria for defining fluvial fans, but because criteria can vary, this paper needs to specifically define it here.

On the same note, how are the wave- and tide-influenced river-dominated deltas actually distinguished (line 366, for example)? Please clarify these criteria as well in the methods.

*Bifurcation vs avulsion terminology*

This is only a matter of wording and so is less important than my previous two comments, but I think the way the authors have defined "bifurcation" as a process of channel splitting driven by mouth bar formation (i.e., line 139) is too narrow and leads to some confusion throughout the paper. Many geomorphologists/sedimentologists, myself included, think of bifurcation as a channel split which can occur via many mechanisms, including avulsion. I think of the great Slingerland and Smith (2004) paper about avulsions – there is a wonderful section in that paper that thinks about how avulsions occur via a bifurcation stability analysis, as just one example of a key reference where avulsions are treated as bifurcations.

In this manuscript, the authors clearly lay out their narrower, mouth-bar focused definition of bifurcation in lines 138-143, so I do understand what they mean. However, is this likely confusion for some readers necessary? Why not call the "bifurcation" group mouth bar bifurcations? That term has process in it, which makes it more equivalent to the "avulsion" category, which is also a process. The word bifurcation is too geometric and isn't tied to a specific process by broad definition.

Changing this terminology would require editing uses of the narrowly defined "bifurcation" throughout the paper & figures, but I think it would be worth it to improve clarity.

Line-specific comments

Line 22-24: the abstract should have more of the actual results in it, including the different bifurcation angles found for deltas vs fans

Line 141: needs older citations defining channel avulsion

Lines 156-165: this paragraph should cite Brooke et al. 2022 and engage with the findings therein about where avulsions occur on deltas vs fluvial fans. In fact I think the ideas from this paper would be useful in other parts of this manuscript as well (such as in section 2.2).

Line 198-207: Cite & consider local vs regional avulsion ideas in Slingerland and Smith (2004), which also has important discussion about avulsion bifurcations that could be useful throughout this manuscript

Lines 261-264: How do you distinguish between active and abandoned channels? Is that distinction important for this? What about splay channels versus main channels? Does it matter if the avulsion bifurcation is partial? This question also came up for me in lines 325-326, that clarity is needed on how splay channels are considered (in both fan and delta environments – since deltas also have splays and they don't form via the mouth bar bifurcation process)

Line 308: typo/wording

Line 326: None of the systems have seasonal change in discharge? That doesn't seem possible…. Probably just a wording issue for this statement

Line 336: typo/wording

Figure 7d: labels beneath the violin plots are overlapping and hard to read

Line 388-389: why is there a discrepancy with Hartley et al. (2010)? Explain/justify

Figure 8: because the key goal of the paper is to compare deltas vs fluvial fans, I do not think these data are presented in the optimal way to make that comparison. It would be easier to compare if the y-axes on width plots were the same for deltas and fans, and same for the length plot y-axes. Additionally, move the width plots in adjacent rows so that it is easy to compare between deltas and fans, and then have the bottom two rows be the length plots for easy comparison.

Lines 470-476: does this slope-related assumption pan out with the data? This would be a good hypothesis to measure and test, because showing the mechanism seems pretty key & important. If the mean down-fan slopes are known for selected fans, it wouldn't be too cumbersome to plot slope vs bifurcation angle

Line 503: for this to be used to distinguish fans vs deltas in seismic datasets, what is the minimum number of measurements you would need to be able to make a conclusion? The means are somewhat different, but there is quite a bit of overlapping range in angle measurements. Can you write more about the required dataset size?

Lines 517-520: cite & incorporate findings from Brooke et al. 2022

Lines 522-523: sediment delivery would not be affected? I'm not sure what you mean by that or how that is related to findings from this paper

---

## Author Comment (AC1)

**Major comments**

1. The treatment of river and tide dominated deltas is confusing. On L186 you say that we neglect W- and T- dominated, but then you do treat them in Fig. 7a. You state that you looked at quantitative metrics for process dominance (L300), but don't give details of any cutoff (e.g. 50% river dominated). I don't mind if you didn't go this specific: one of my key takeaways is that the process regime doesn't particularly influence the angle, which is nice and a worthy paper conclusion. Then in L371 it sounds like wave dominated deltas were omitted from the statistics.

We understand some of the confusion regarding wave- and tide-dominated deltas and have addressed these in the text. To clarify our terminology, we elaborate further on our delta classifications in section 3.4 distinguishing wave- and tide- "dominated" and "influenced" deltas. Wave-dominated deltas (e.g., São Francisco, Eel), are characterized by strandplanes and a complete absence of bifurcations. Wave-influenced deltas retain features such as strandplains but exhibit clear, measurable channel bifurcations. Similarly, tide-dominated deltas (e.g., Fly, Yangtze) have a limited number of channels that widen substantially seaward, whereas tide-influenced deltas show channel widening in the most distal channels. We combined these morphological criteria with established classifications (Broaddus et al., 2022; Galloway, 1975; Nienhuis et al., 2015, 2018; Paniagua-Arroyave & Nienhuis, 2025; Vulis et al., 2023) to categorize the deltas included in our analysis. All classifications and measurements for individual deltas are provided in the Supplementary Information, enabling this dataset to be reused and reinterpreted under alternative delta classification schemes.

2. L253, Several aspects of network mapping need to be described better.  Particularly, is there a lower limit of channel width that you stop measuring at? Do short side branches or tie channels count as channels? I don't mean for you to be endlessly bogged down in these choices, but it would be good to include your choices and shapefiles as a supplement so that your work can be introduced.

We have clarified in the revised text in section 3.1 that channel mapping begins at the first observable branching point and includes all channel segments that either enter a body of water or terminate on land. Tie channels were generally excluded, as most delta channel networks in our dataset discharge into open water (e.g., Fig. 1). In rare cases where tie channels were morphologically indistinguishable from distributary channels, they may be included.

3. L393 I don't believe that Normalized Channel Length is defined anywhere. Is it a channel length divided by apex channel width? Average channel width of the reach? It might be interesting to cite Jerolmack (2009), which does the same normalization.

Thank you for identifying this oversight. We now explicitly define normalized channel length as channel length divided by the initial first-order channel width, following the methodology of Edmonds and Slingerland (2007). To further clarify and contextualize this

approach, we have also added a citation to Jerolmack (2009), which applies a similar normalization method. This definition and supporting references are now included in Section 3.2, alongside our description of normalized channel width values.

4. In section 5.1 you discuss how small angles are more likely to be from a fan than a delta. Are you arguing that 60 or 64 degree average angle is a good cutoff for fans vs deltas? A number like this would be really helpful, but I think it requires some statistical modeling beyond what is here. For example, if you had just 2 measurements on an ancient system and their average was 66 degrees, you wouldn't have much confidence. I'm not requiring more analysis, but this is an important passage and as yet we don't have good guidelines for applying this work.

We agree that Section 5.1 was lacking in clear guidance for how our results could be applied to future work. In the revised manuscript, we have clarified that while our dataset shows that smaller angles are generally more likely to occur in fluvial fans compared to deltas, we do not intend to suggest a strict numerical cutoff (e.g., 60–64°) for distinguishing between the two systems. We now emphasize that reliable application of this metric requires sufficient sample sizes (n ≥ 10) and should be interpreted in combination with other geomorphic indicators such as normalized channel length values and trends in angle sizes changes basinward. These revisions highlight the utility of our findings while also cautioning against oversimplification.

**Minor comments**
5. L22, fluvial fans and deltas have distinguished before (Van Dijk et al., 2009). Perhaps specify that you are looking at quantitative differences in channel network morphometrics.

The manuscript now specifies that this study is an initial attempt to compare and distinguish deltas and fluvial fans using quantitative channel network morphometrics.

6. L144 "Deltas always form where the mouth of a river enters a body of water." This is not an essential point to the paper, but it is easy to find counterexamples to this, such as the Amazon. Perhaps change always to generally.

The sentence has been revised to clarify that deltas generally form along standing bodies of water only at places where rivers enter, accounting for cases such as the Amazon River delta.

7. L179-182 While E&S did propose this reason for channel length reduction, I do not think that it is broadly confirmed. I find that your empirical results are interesting, but that normalized channel lengths of 100s and 1000s are far too long to be explained by momentum. My point here is to just state that the proposed mechanisms have not been firmly established yet, and your certainty overstates the case.

We have reworded the text in this sentence to indicate this uncertainty. However we do not attempt to link the decrease in channel length and widths from Edmonds and Slingerland, (2007) to the trends observed in fluvial fan channel networks. This section only concerns deltas.

8. L245 remove existing.

The word "existing" has been removed from the text.

9. L262 By "not considering channels" I think you mean that you measure them but don't have them influence the channel ordering, right?

We did not include these channels in our measurements, nor did they influence channel ordering. We have replaced "consider" with "map or measure" to clarify this point.

10. L287-288 branching into three channels is sometimes called a "furcation" (Shaw et al., 2018) or a polyfurcation (Chamberlain et al., 2018).

We have incorporated these alternative terminologies, "furcation" and "polyfurcation," along with the corresponding references, into the revised manuscript.

11. L308, confusing sentence structure.

We have reworked these sentences to clearly convey the content of the database and provide clearer examples of fan termination styles.

12. L321, Python- and Pandas- readable.

We have corrected the capitalization and punctuation for these two words.

13. L451-452 this observation is consistent with Coffey and Shaw.

We have now included text describing that differences in channel reach angle measurements are also consistent with findings from Coffey and Show, (2017).

14. L459, I think "diffusion in unchannelized flow" could be improved. Perhaps, "flow patterns at channel tips well-explained by diffusive processes.

We have revised the sentence to specify that flow patterns at channel tips are well-explained by diffusive processes.

15. Our recent paper (Shaw et al., 2025) shows that many large systems have both proximal fan and distal delta platform components. Could this possibly explain why angles tend to be larger at the higher order channels at the distal end of fans?

We believe, fluvial fan and delta are defined somewhat differently in our paper as compared to Shaw et al. (2025). We define these landforms by formative processes, rather than by gradient as in Shaw et al. (2025).In our study, comparisons between gradient and mean angles (Supplementary Information) do not show any significant correlations. This may be because the overall landform gradient is not an accurate representation of the slope of fluvial fans, which have been shown to decrease in gradient downfan even when they are completely terrestrial and do not enter a basin (e.g., Chakraborty et al., 2010). Some landforms in our dataset contain fluvial fans that transition into river deltas, either close to the fan (e.g., Saskatchewan) or much further downstream (e.g., the Niger). In these cases, the fan and delta portions display significantly different mean angles (Fig. 5a and 5b). The bimodal distribution of higher order fluvial fan channels around 90° we argue is indicative of crevasse channel development, which could explain the increase of larger angles in these groups.

---

## Author Comment (AC2)

**Main concerns & questions**

1. River deltas versus water-terminating fluvial fans. The distinction between river deltas and fluvial fans is key to the premise of this paper, which asserts that these are two fundamentally different landforms with different bifurcation processes and thus different morphologies. Overall this makes sense to me, except in the case that fluvial fans terminate in standing bodies of water (i.e., oceans and lakes). If the distributary fluvial network ends in a standing body of water, isn't that simply a river delta? Lines 74-84 mention this ambiguity, but I don't understand how this is resolved in the paper. At a minimum, more text is needed to explain how some landforms that end in water are classified as fluvial fans, and others are classified as river deltas. Figure 7d shows the different terminations for the fluvial fans in this study (although not the n for each one), and many of them are in the "lake" and "marine" categories. How are those distinct from river deltas? Addressing this question may be a matter of revision to the writing to better explain in advance how water-terminating fans differ from river deltas (i.e., spell this out clearly in line 74-84, using criteria that aren't the ones being tested in this study (bifurcation angle, etc.)). However, if such independent criteria don't exist, then I wonder if it is necessary to remove the lake- and marine-terminating fans from this analysis. Having two independent populations is very important in this comparative analysis, so I consider this to be a critical issue that needs to be addressed before publication.

We understand the confusion regarding fluvial fans that terminate into standing bodies of water. In fact, this confusion is a key motivation for this work, as deltas and fluvial fans are likely to respond differently to climate change and urbanization.

As discussed in the Introduction and Sections 2.1 and 2.2, river deltas and fluvial fans form via different sedimentological processes, which generate distinct channel network morphometrics. We modified the text in the introduction in attempt to further highlight these differences.

Several landforms included in this study (e.g., the Saskatchewan and Niger fluvial fans and deltas) have both a fluvial fan and a downstream river delta, and our analysis shows that these features cluster distinctly into fan and delta categories. Regarding Figure 7d, we now clarify in Section 4.1 that all fan termination styles are represented by 4–6 landforms each, providing clearer context for water-terminating fans within the dataset. Also, we test for differences in fluvial fan channel network angles by termination style, and there are no statistical differences in lake and ocean terminating fans vs other termination styles.

2. River vs wave vs tide delta criteria. The matter of defining the type of delta (river vs wave vs tide) is unclear in this manuscript and needs clarification or to be refined with more quantitative criteria, if no quantitative metrics were applied before sites were chosen. In the intro (lines 187-188), the authors specify that only river-dominated deltas were used, but it is unclear how this is established and also there

are results from wave- and tide-dominated deltas later on (e.g., Figure 9). The current methods sentence (lines 298-299) is definitely not specific enough about distinguishing between types. I do think the cited literature in this sentence is the appropriate body of work to establish specific criteria for defining fluvial fans, but because criteria can vary, this paper needs to specifically define it here. On the same note, how are the wave- and tide-influenced river-dominated deltas actually distinguished (line 366, for example)? Please clarify these criteria as well in the methods.

We acknowledge the need for clearer definitions for delta classification and have revised the text accordingly. In Section 3.4, we now explicitly define our criteria for distinguishing wave- and tide-dominated versus wave- and tide-influenced river deltas. Wave-dominated deltas (e.g., São Francisco, Eel) are characterized by strandplains and a complete absence of bifurcations, whereas wave-influenced deltas retain strandplains but exhibit clear, measurable channel bifurcations. Tide-dominated deltas (e.g., Fly, Yangtze) feature a limited number of channels that widen substantially seaward, while tide-influenced deltas show widening only in their most distal channels. The Supplementary Data which we now reference here includes data on our classification of delta types.

There are no results from wave- or tide-dominated deltas included in this manuscript. We only used wave- and tide-influenced (river-dominated) deltas, including the display in Figure 7 (there is no Figure 9).

3. Bifurcation vs avulsion terminology. This is only a matter of wording and so is less important than my previous two comments, but I think the way the authors have defined "bifurcation" as a process of channel splitting driven by mouth bar formation (i.e., line 139) is too narrow and leads to some confusion throughout the paper. Many geomorphologists/sedimentologists, myself included, think of bifurcation as a channel split which can occur via many mechanisms, including avulsion. I think of the great Slingerland and Smith (2004) paper about avulsions – there is a wonderful section in that paper that thinks about how avulsions occur via a bifurcation stability analysis, as just one example of a key reference where avulsions are treated as bifurcations. In this manuscript, the authors clearly lay out their narrower, mouth-bar focused definition of bifurcation in lines 138-143, so I do understand what they mean. However, is this likely confusion for some readers necessary? Why not call the "bifurcation" group mouth bar bifurcations? That term has process in it, which makes it more equivalent to the "avulsion" category, which is also a process. The word bifurcation is too geometric and isn't tied to a specific process by broad definition. Changing this terminology would require editing uses of the narrowly defined "bifurcation" throughout the paper & figures, but I think it would be worth it to improve clarity.

We appreciate your comments regarding our terminology for bifurcation versus avulsion and agree that these terms are differently in the community. Consequently, whichever way we proceed, some community members will be confused. We chose to define bifurcation

more narrowly as channel splitting driven by mouth-bar deposition, as this usage is common in the deltaic literature, and helps us to be clear about the differences in formative processes in deltas and fluvial fans. We acknowledge that this narrower definition may not capture the full spectrum of processes encompassed by bifurcation in the broader geomorphic sense, but we felt it necessary to distinguish between mouth bar–driven bifurcations and avulsion-driven channel creation to reduce ambiguity in how these processes are expressed in fan-shaped landforms. The lack of standardized terminology in the literature has indeed contributed to confusion regarding the processes that govern channel network evolution on deltas and fluvial fans. By explicitly defining bifurcation in our study, we aim to provide clarity for our classification framework. We modified the text in the introduction to more explicitly state the narrower process-based use of "bifurcation".

**Line-specific comments**

4. Line 22-24: the abstract should have more of the actual results in it, including the different bifurcation angles found for deltas vs fans.

We have added the mean bifurcation angle values for deltas and fluvial fans to the abstract, as these are a key diagnostic criterion in the study.

5. Line 141: needs older citations defining channel avulsion.

We have added additional older references that define channel avulsion to strengthen the background for our manuscript.

6. Lines 156-165: this paragraph should cite Brooke et al. 2022 and engage with the findings therein about where avulsions occur on deltas vs fluvial fans. In fact I think the ideas from this paper would be useful in other parts of this manuscript as well (such as in section 2.2).

We have further integrated the findings of Brooke et al., (2022) in sections 2.1 and 2.2 to better discuss and differentiate avulsions on deltas and fluvial fans.

7. Line 198-207: Cite & consider local vs regional avulsion ideas in Slingerland and Smith (2004), which also has important discussion about avulsion bifurcations that could be useful throughout this manuscript.

We have added discussion of regional avulsions in Section 2.2, citing Slingerland and Smith (2004). Additionally, we included a note on local avulsions in Section 3.1 (Methods), clarifying that these typically rejoin the downstream channel, and are therefore not included in our channel network measurements.

8. Lines 261-264: How do you distinguish between active and abandoned channels? Is that distinction important for this? What about splay channels versus main

channels? Does it matter if the avulsion bifurcation is partial? This question also came up for me in lines 325-326, that clarity is needed on how splay channels are considered (in both fan and delta environments – since deltas also have splays and they don't form via the mouth bar bifurcation process).

We recognize the ambiguity in distinguishing active versus abandoned channels. Section 3.1 now clarifies how paleochannels are identified and why they are included. For partial avulsions, both older and newer channels may convey flow simultaneously, but this does not affect our methodology as long as channels do not merge downstream. Splay channels are included in some measurements, with their occurrence discussed in Section 5.1. We also emphasize that a sufficiently large sample size is essential to capture representative bifurcation angles and reduce the influence of splays or other local anomalies on mean values for a given fluvial fan or delta.

9. Line 308: typo/wording

We have rewritten the text here to provide more clarity in the terminologies and address the type.

10. Line 326: None of the systems have seasonal change in discharge? That doesn't seem possible…. Probably just a wording issue for this statement.

The statement was intended to convey that, across the datasets we examined, we do not observe significant changes in channel width or discharge across individual fans that could be attributed to differences in the timing of image capture relative to precipitation events. Fluvial fans are highly sensitive to precipitation and short-term variability is possible. Because fluvial fans can extend for hundreds of kilometers, satellite images are often mosaicked from multiple acquisitions taken at different times. As a result, we do not see systematic differences in channel activity across the fan (e.g., lower fan channels do not consistently appear wider in one area of the fan due to recent precipitation).

11. Line 336: typo/wording

We have omitted the typo in this sentence.

12. Figure 7d: labels beneath the violin plots are overlapping and hard to read

We increased the x-axis label sizes on the violin plots in Figure 7 to improve readability, with specific adjustments on Figure 7d for clarity.

13. Line 388-389: why is there a discrepancy with Hartley et al. (2010)? Explain/justify

We have addressed the discrepancies with Hartley et al. (2010) by providing examples in section 4.1 where their definitions do not align with the observed environments, such as referring to playa fans as lacustrine or ocean fans as contributory.

14. Figure 8: because the key goal of the paper is to compare deltas vs fluvial fans, I do not think these data are presented in the optimal way to make that comparison. It would be easier to compare if the y-axes on width plots were the same for deltas and fans, and same for the length plot y-axes. Additionally, move the width plots in adjacent rows so that it is easy to compare between deltas and fans, and then have the bottom two rows be the length plots for easy comparison.

We appreciate the suggestion to standardize the y-axis scales for easier comparison between deltas and fluvial fans. We have applied consistent y-axis scales for the width plots. For normalized channel length, standardizing the scale caused boxes beyond channel order 5 in fans to become unreadable, so we retained the original scales but have clearly highlighted the differences on the figure axes in the legend. We also reorganized the plot to place widths and lengths on the same rows to improve comparison between the plots.

15. Lines 470-476: does this slope-related assumption pan out with the data? This would be a good hypothesis to measure and test, because showing the mechanism seems pretty key & important. If the mean down-fan slopes are known for selected fans, it wouldn't be too cumbersome to plot slope vs bifurcation angle

We do not observe any trends in relation to average angle with respect to fan gradient (Supplementary Figure 2). However, as noted in our results section, Fig. 6b shows that mean angle increases with channel order in fans, which can be considered an indirect measurement of how dowfan a channel segment is. One potential cause of the discrepancy is that gradients over fans can change from steeper in proximal to significantly lower in more distal regions of fans (e.g. Chakraborty et al., 2010), and that a mean angle does not reflect to changing gradient of the fans. We have now included text in discussion 5.2 to recommend future research directions involving

16. Line 503: for this to be used to distinguish fans vs deltas in seismic datasets, what is the minimum number of measurements you would need to be able to make a conclusion? The means are somewhat different, but there is quite a bit of overlapping range in angle measurements. Can you write more about the required dataset size?

In their study of branching angles in the seismic record, Mahon et al., (2024) use as little as one or two observed measurements for their interpretations. As stated in our discussion 5.1, low sample sizes can lead to varied mean angles which can lead to inconclusive result. We recommend a robust sample size. We now highlight that a greater amount of

measurements (approximately equal to or greater than 10) is necessary help to more accurately constrain the mean branching angle.

17. Lines 517-520: cite & incorporate findings from Brooke et al. 2022

We have now incorporated and cited the findings in the discussion section 5.4.

18. Lines 522-523: sediment delivery would not be affected? I'm not sure what you mean by that or how that is related to findings from this paper

We have clarified our point to specify that sedimentation would not be affected by sea level rise across most of a fluvial fans surface except in areas near fan toe.

---

## Author Comment (AC3)

1. **Major considerations**: The choice of metrics is my main concern. Restricting the analysis to bifurcation angle and downstream channel width/order feels too narrow. Other metrics such as lateral channel mobility or avulsion frequency would provide a richer basis for comparison and may lead to more meaningful separation. When a fan or delta terminates in standing water, downstream boundary effects become critical, and prior work (e.g., Carlson et al., 2018; Wang et al., 2019) shows that boundary conditions strongly influence channel number, lateral migration rate, and sediment bypass. These findings should inform the interpretation here. Even if channel depth cannot be measured from imagery, alternatives such as migration rates or wetted frequency maps (e.g., Piliouras et al., 2017) could help test whether channel dynamics differ between deltas and fluvial fans. This would require using data from more than a single snapshot in time. The authors are working on scales where existing channel metric tools could be applied, and doing so would provide a clearer picture of system dynamics across time and discharge conditions. In general, the paper would be stronger if the classification were less prescriptive. Rather than setting a framework in advance, I would like to see populations emerge from the data, and then understand when a fluvial fan behaves like a delta and when it does not. That approach would make for a more compelling contribution.

We thank the reviewer for the suggestion to incorporate additional metrics such as lateral mobility, avulsion frequency, or wetted frequency. We agree that these parameters as defined in Carlson et al., (2018) and Wang et al., (2019) would provide valuable insight into delta dynamics; however, they can require either multi-temporal or bathymetric datasets and, nor are they easily mappable morphometric criteria that can be applied to both delta and fluvial fan channel networks – especially since many fluvial fans terminate in terrestrial environment. Our intent is to focus on branching angles and downstream channel width/length because these metrics can be consistently extracted from single high-resolution images and compared across a wide range of environments (both terrestrial and marine).

**Minor considerations**

2. The selection of case studies is unclear. How is the threshold for fluvially dominated deltas quantified?

We understand some of the ambiguity in defining our definitions from wave- and tide-influences and dominated deltas. We have addressed your comments by clarifying our classifications in section 3.4 to more accurately specify our methodology for distinguishing wave- and tide- "dominated" and "influenced" deltas. Wave-dominated deltas are characterized by strandplanes and a complete absence of bifurcations, whereas wave-influenced deltas retain features such as strandplains but exhibit clear, measurable channel bifurcations. Similarly, tide-dominated deltas have a limited number of channels that widen substantially seaward, unlike tide-influenced deltas show channel widening only in the most distal channels. We have also now included references to our Supplementary Data which contains the assigned delta classification as well.

3. Should confinement be considered as a control?

We did not consider confinement because confinement is often difficult to quantify consistently across global datasets. Shaw et al., (2025) attempt to quantify confinement in their study of deltas, however some of confinement angles are inconsistent with observed topographic data. Moreover, we wanted to restrict our metrics to existing morphometric controls such bifurcation angle (Coffey and Shaw, 2017) and channel length and width trends (Edmonds and Slingerland, 2007; Jerolmack, 2009). Future targeted studies on the controls of confinement on channel network development would be interesting; for instance existing fluvial fan models (e.g. Harrison and Edmonds, 2023) typically consider no confinement.

4. Figure 2 needs clearer labeling (e.g., upstream/downstream).

We intended for our mapping symbology using a combination of color palette and decreasing line thickness to indicate downfan morphology, however we recognize that there may by some confusion for downstream directions especially for terrestrial fans. We have included white arrows in the figure that point from the fan apex to the downfan direction.

5. Several figures would benefit from improved color schemes and alternative presentation. Scatter plots, for example, could more clearly show whether two populations emerge.

For the channel ordering figures, we selected a ROYGBIV color scheme with bold, high-contrast colors to maximize clarity and accessibility, particularly for color-blind readers. While other studies mapping delta channel networks (e.g., Dong et al., 2016; 2020) have used alternative color schemes, we found those less interpretable under accessibility considerations. In addition to color, we emphasized channel hierarchy by scaling line thickness: lower-order channels (order 1) are drawn thickest, while higher-order, distal channels are progressively thinner. We believe this dual coding (color and thickness) improves interpretability and reduces reliance on color perception alone. We are definitely open for making scatter plots and specific suggestions here would be appreciated.

6. The GitHub link provided does not work.

We have reviewed the GitHub link, and it does work.